# Juvenile Membranous Nephropathy Developed after Human Papillomavirus (HPV) Vaccination

**DOI:** 10.3390/vaccines10091442

**Published:** 2022-09-01

**Authors:** Haruna Arakawa, Shohei Yokoyama, Takehiro Ohira, Dedong Kang, Kazuho Honda, Yoshihiko Ueda, Akihiro Tojo

**Affiliations:** 1Department of Nephrology & Hypertension, Dokkyo Medical University, Mibu 321-0293, Japan; 2Department of Anatomy, Showa University School of Medicine, Tokyo 142-8555, Japan; 3Department of Pathology, Dokkyo Medical University Saitama Medical Center, Saitama 343-8555, Japan

**Keywords:** human papillomavirus, vaccination, membranous nephropathy, p16 INK4a, mass spectrometry

## Abstract

A 16-year-old girl with no history of renal disease had a fever of 38 °C after her second HPV vaccination and was identified as positive for proteinuria. As she maintained urinary protein of 3.10 g/gCr and 5–9 urinary red blood cells/HPF, a renal biopsy was performed and small spikes on PAM staining with the granular deposition of IgG1++ and IgG3+ on the glomerular capillary wall were discovered by immunofluorescence, although PLA2R immunostaining was negative. Analysis by electron microscope showed electron density deposition in the form of fine particles under the epithelium. The diagnosis was secondary membranous nephropathy stage II. Immunostaining with the anti-p16 INK4a antibody was positive for glomerular cells, and Western blot analysis of urinary protein showed a positive band for p16 INK4a. However, laser-microdissection mass spectrometry analysis of a paraffin section of glomeruli failed to detect HPV proteins. It is possible that the patient was already infected with HPV and administration of the HPV vaccine may have caused secondary membranous nephropathy.

## 1. Introduction

SARS-CoV-2 vaccination was promoted during the COVID-19 pandemic; however, the effectiveness and side effects of the mRNA vaccines for SARS-CoV-2 were not fully elucidated. Gross hematuria in patients with IgA nephropathy and recurrence of nephrotic syndrome in patients with minimal change nephrotic syndrome (MCNS) were reported as side effects following SARS-CoV-2 vaccination [1,2,3,4]. Some young people are concerned about adverse reactions to the SARS-CoV-2 vaccination, especially in Japan, where young people have a distrust of the vaccine due to cases of side effects caused by the HPV vaccine [5].

Human papillomavirus HPV is a double-stranded DNA virus that can be divided into more than 100 types from the gene sequence of the surface capsid protein L1. High-risk HPV16, 18, 31, 33, 45, 51, 52, 56, 58, and 66 cause cervical cancer, and low-risk HPV6, 11, etc., cause condyloma acuminata [6,7,8]. In Japan, the recombinant divalent human papillomavirus vaccine Cervarix for high-risk HPV 16/18 was sold in 2009, and regular vaccination started in 2013 for 6th-grade elementary school to 1st-grade middle high school girls [8]. However, Cervarix was discontinued due to reports of adverse reactions, including chronic pain, motor impairment, and other symptoms after HPV vaccination. The recombinant precipitated 4-valent HPV-like particle vaccine has been on the market since 2011, and the recombinant precipitated 9-valent HPV-like particle vaccine has been on the market since 2021, although the Ministry of Health, Labor, and Welfare in Japan announced a suspension of the proactive recommendation for routine use of the HPV vaccine in the national immunization program in June 2013 [9] until November 2021. In Cervarix, the capsid protein of the recombinant precipitated divalent human papillomavirus is atomized and aluminum hydroxide and monophosphoryl lipid A (MPL) derived from the cell membrane of Salmonella are used as an adjuvant. These adjuvants destroy cells at the administration site, and the destroyed autologous cell DNA and proteins are recognized by the DAMPs in the innate immune response system [10] and cause not only fever and local swelling but also Guillain–Barré syndrome (GBS), systemic lupus erythematosus (SLE), autoimmune hepatitis, and cerebral vasculitis [11]. We experienced a case of secondary membranous nephropathy after the second injection of the divalent HPV vaccine Cervarix, and here we discuss the mechanism of membranous nephropathy after HPV vaccination.

## 2. Case Presentation

A 16-year-old girl had a fever of 39.0 °C for 4–5 days after the second HPV vaccination, even though she had no fever following the first HPV vaccination one month prior. She was prescribed acetaminophen, and she developed proteinuria for the first time 5 days after receiving the second vaccination. Her fever was resolved spontaneously, and proteinuria was not checked until the next school year in a urine checkup, which showed proteinuria 3+ and occult blood. She was hospitalized for examination by renal biopsy. She had a history of allergy to pollen. Her family history included cerebral hemorrhage in her maternal great-grandfather, multiple system atrophy in her grandfather, and hypertension in her grandmother. Physical findings on admission were blood pressure 120/82 mmHg, heart rate 115 bpm, body temperature 36.2 °C, SpO_2_ 98% (room air) without tonsillitis, lymphadenopathy, or neurological dysfunction. A urinalysis showed urinary protein, 1.46 g/gCr, 5–9 RBC/HPF, 1–4 WBC/HPF, cast (-), NAG 2.7 U/L, and selectivity index 0.121. A blood test showed WBC 8100 × 10^6^/L, (neutrophil 68.2%, eosinophil 1.7%, basophil 0.7%, monocyte 8.3%, lymphocyte 21.1%), Hb11.9 g/dL, and platelet 39.8 × 10^10^/L. Biochemistry data revealed TP 5.5 g/dL, Alb 3.5 g/dL, UN 11.0 mg/dL, Cr 0.47 mg/dL, eGFR 147.8 mL/min/1.73 m^2^, and CRP 0.01 mg/dL. Immunological tests found ANA (-), anti-dsDNA antibody (-), anti-SM antibody (-), IgG 544 mg/dL, IgA 112.4 mg/dL, IgM 134.7 mg/dL, IgE 34.3 mg/dL, C_3_ 141.9 mg/dL, C_4_ 23.2 mg/dL, ASO 14 IU/mL, lupus anticoagulant negative, anti-CLβ2GP1 antibody 1.2 U/mL, anti-SS A/B antibody (−/−), MPO-ANCA (-), PR3-ANCA (-), anti-GBM antibody (-), HBs antigen (-), HCV antibody (-), and D dimer 0.3μg/mL. These data indicate that there were no possibilities of lupus nephritis, other autoimmune diseases, or antiphospholipid syndrome.

## 3. Renal Biopsy and Laser-Microdissection Mass Spectrometry

### 3.1. Renal Biopsy

Two samples of cortex were collected, including a total of 25 glomeruli without global sclerosis, mild mesangial cell proliferation in 10 glomeruli (40%), and normal tubulointerstitium and vascular system. PAM staining showed mild spikes with subepithelial immune complexes by AZAN staining, which were granularly stained along the capillary wall by fluorescent immunostaining for IgG (Figure 1).

Among the IgG subclasses, IgG1 was the strongest, IgG3 was mildly stained (Figure 2), and immunofluorescence for phospholipase A2 receptor (PLA2R) was negative (Figure 3A), suggesting secondary membranous nephropathy. Electron microscopy revealed an electron-dense deposit in the subepithelial membrane and partly in the basement membrane, indicating stage II–III membranous nephropathy (Figure 4).

### 3.2. Surrogate Marker for HPV Infection

Immunohistochemistry was performed with the Leica auto-immune stain system using the antibody against the anti-p16-INK4a antibody (CINtec^®^ Histology, Roche Diagnostics KK, Tokyo, Japan), which is typically used as a surrogate marker of HPV infection in cervical cancer and oropharyngeal cancer [12,13]. The anti-p16-INK4a antibody showed significant staining in the intra-glomerular cells (Figure 3B).

Furthermore, Western blot analysis identified p16 protein in the urinary protein at the time of renal biopsy. On the other hand, it was not found in the urine of patients with secondary membranous nephropathy related to cancer or primary membranous nephropathy—it was specific to this case (Figure 5). Therefore, it was suggested that this case could have already been infected with HPV before the time of renal biopsy.

### 3.3. Laser Microdissection Mass Spectrometry

We performed laser microdissection mass spectrometry (LMD-MS), as mentioned previously [14], to detect HPV viral capsule proteins. LMD-MS failed to detect HPV protein or antigens for membranous nephropathy, including THSD7A, EXT1/2, NELL1, Sema 3B, PCDH7, HTRA1, Contactin 1 [15,16], which were negative, except for a small amount of PLA2R in one sample (Table 1). The cytokeratin-related proteins were increased, whereas podocyte proteins such as nephrin, podocin, and podocalyxin were decreased compared to the control glomeruli from the renal transplantation at 1 h biopsy (Table 1, Appendix A).

### 3.4. Clinical Course

The angiotensin receptor blocker losartan 25 mg was administered to reduce the urinary protein, decreasing it from 3.1 g /gCr to 0.19 to 0.65 g /gCr. When the dose of losartan was reduced to 12.5 mg after 6 months, urinary protein increased slightly to 0.41 g/gCr (Figure 6).

## 4. Discussion

This study focuses on a report of membranous nephropathy in an adolescent girl after HPV vaccination, and the pathogenic mechanism is investigated.

### 4.1. Characteristics of Adolescent Membranous Nephropathy

Membranous nephropathy is a major cause of nephrotic syndrome in middle-aged and elderly people but is rare in adolescents (1–2% of renal biopsies) [17]. Histopathologically, adolescent membranous nephropathy is found in a relatively early phase of Ehrenreich–Churg classification stages 1 and 2, in which patients are frequently positive for IgA, IgM, and C1q, in addition to IgG and C3 [18]. In the IgG subclass, IgG4 was 87.5%, whereas IgG1 was 46.9% and IgG3 was 56.3% were positive in half of the cases [19], and the frequency of PLA2R positivity was slightly lower than that in adults [20]. These data indicate that it is important to investigate secondary membranous nephropathy, such as SLE and HBV infection, in adolescent cases [18]. The present case was secondary membranous nephropathy with predominant IgG1 and IgG3 deposition but was negative for PLA2R. Most of the well-known causes of secondary membranous nephropathy were ruled out from clinical findings and laboratory tests. Therefore, HPV vaccination is presumed to be the cause of membranous nephropathy.

### 4.2. HPV Virus Infection and Kidney Disease

HPV is a sexually transmitted disease that is persistently transmitted locally to the cervix, and high-risk HPV types 16 and 18 cause about 70% of cervical cancers [8]. Interestingly, mother-to-child transmission can occur at birth [21], and 22.8% of newborns have HPV detected in the oral cavity at birth [22]. It is possible that even young people who have never had sexual intercourse have already been infected with HPV. In addition, 30.3% of 122 patients with renal cell carcinoma were found to be positive for HPV-DNA in the renal cell carcinoma site of the paraffin specimens by PCR method, 20.3% were found to be positive by immunostaining for p16-INK4a, and 45% were found to be positive for HPV by the in situ hybridization method [23]. This indicates that HPV was infected and latent in the kidney. In this case, p16-INK4a protein, a surrogate marker of HPV infection [12,13], was found in glomerular cells (Figure 4B) by immunohistochemistry and also in the urinary protein by Western blotting analysis (Figure 5). Therefore, it is possible that the patient was infected with HPV, having an antibody against it, and administration of viral protein as a vaccine formed circulating immune complexes and developed proteinuria soon after vaccination. The p16-INK4a is also detected in the glomerulus and tubules of the aging kidney and in kidneys with chronic allograft rejection [24,25], even though this was not the case with this young patient. HPV envelope proteins, such as E1 proteins, induce overexpression of a set of genes associated with proliferation and differentiation processes and downregulation of immune response genes [26]. LMD-MS, in this case, showed an increase in cytoskeletal proteins and epithelial junctional proteins as well, as downregulation of nephrin, podocin, and podocalyxin may reflect HPV infection in the podocytes. Unfortunately, we could not detect HPV envelope proteins by LMD-MS analysis. The possibility of primary membranous nephropathy still remains, as LMD-MS analysis detected a small amount of PLA2R protein in one glomerular sample.

### 4.3. Kidney Disease associated with Vaccination

In this study, the urinary protein was observed with a fever of 39 °C after two doses of a recombinant precipitated divalent HPV-like particle vaccine.

As shown in Table 2, a case of acute kidney injury and nephrotic syndrome due to membranous nephropathy was reported after influenza vaccination [27]. It has been reported that nephrotic syndrome could be caused by the HBV vaccine [28], pneumococcal vaccine [29], and COVID-19 vaccine [1,30]. However, there has never been a report of nephrotic syndrome caused by the HPV vaccine, and to the best of our knowledge, this study presents the first report of such a case.

The HPV vaccine Cervarix contains added aluminum hydroxide and monophosphoryl lipid A (MPL) derived from the cell membrane of Salmonella as an adjuvant. Aluminum oxyhydroxide (alum) binds to viral DNA fragments and lipid A derivatives to form alum-nanoparticles, which are taken up by macrophages and form a granulomatous lesion called macrophagic myofasciitis (MMF) [40,41]. MMF in the vaccine injection site induces cell death, and adjuvants conjugate with degraded nuclear DNA and proteins, producing autoantibodies, which may cause Guillain–Barré syndrome, SLE, autologous immune hepatitis, and cerebrovascular inflammation. These are so-called adjuvant diseases, an autoimmune/autoinflammatory syndrome induced by adjuvant (ASIA) [40,42,43]. In this case, the possibility of lupus was ruled out from clinical findings and immunological data.

## 5. Conclusions

We reported a case of young-onset IgG1-dominant and PLA2R-negative secondary membranous nephropathy after HPV vaccination. Since p16-INK4a was positive in glomerular and urinary proteins, she may have been infected with HPV, and administration of HPV envelop protein vaccines could be implicated in the development of secondary membranous nephropathy.

## 6. Take-Home Message and Lessons Learned

It is important to check proteinuria after HPV vaccination. Secondary membranous nephropathy could occur after HPV vaccination with viral proteins. If it is possible to check the plasma antibody titers for HPV before vaccination, this may help to prevent the occurrence of membranous nephropathy.

## Figures and Tables

**Figure 1 vaccines-10-01442-f001:**
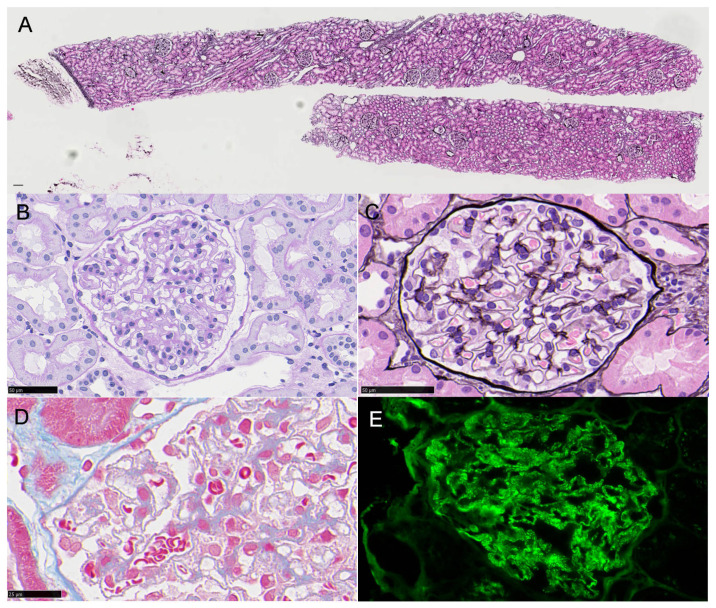
Renal biopsy. PAM staining (**A**,**C**), PAS staining (**B**), Azan staining (**D**), and immunofluorescence of IgG. The bars indicate 50 μm (**B**,**C**,**E**) and 25 μm (**D**).

**Figure 2 vaccines-10-01442-f002:**
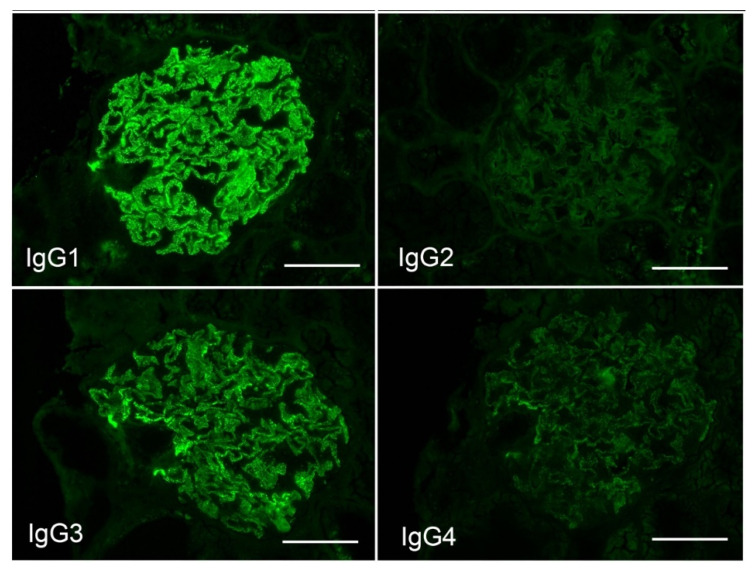
Immunofluorescence of IgG subclass staining. The bars indicate 50 μm.

**Figure 3 vaccines-10-01442-f003:**
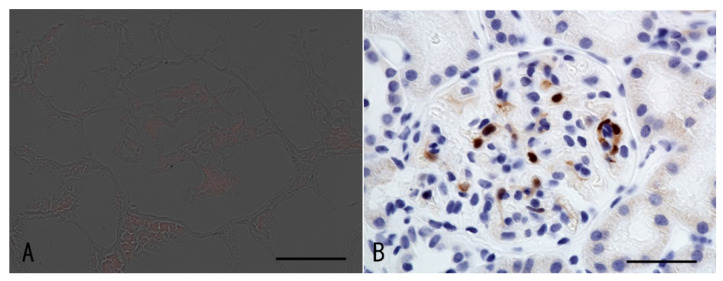
Immunofluorescence of PLA2R (**A**) and immunostaining for the p16-INK4a antibody (**B**). The bars indicate 50 μm.

**Figure 4 vaccines-10-01442-f004:**
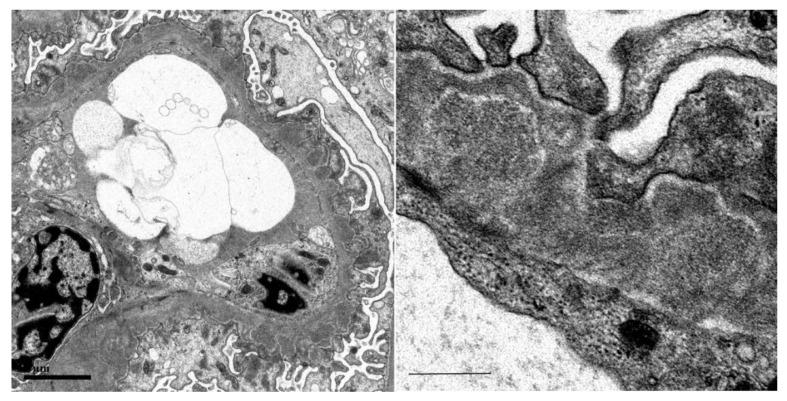
Electron microscopy. The bars indicate 2 μm and 0.5 μm.

**Figure 5 vaccines-10-01442-f005:**
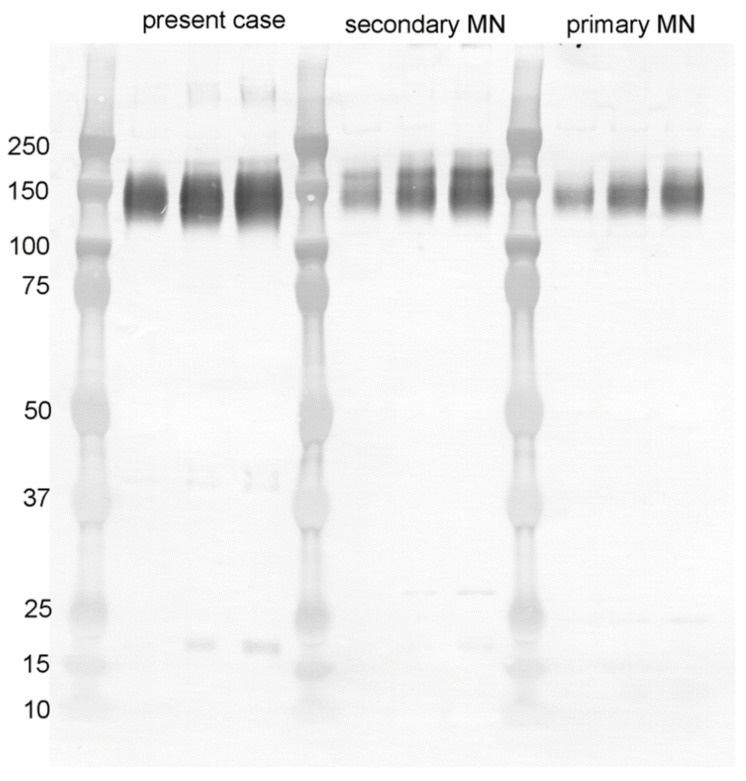
Western blot of the urinary protein at the time of renal biopsy for p16-INK4a. The urine of the present case showed a band at MW16kD, whereas the urine of other cases of membranous nephropathy did not show a band for p16-INK4a.

**Figure 6 vaccines-10-01442-f006:**
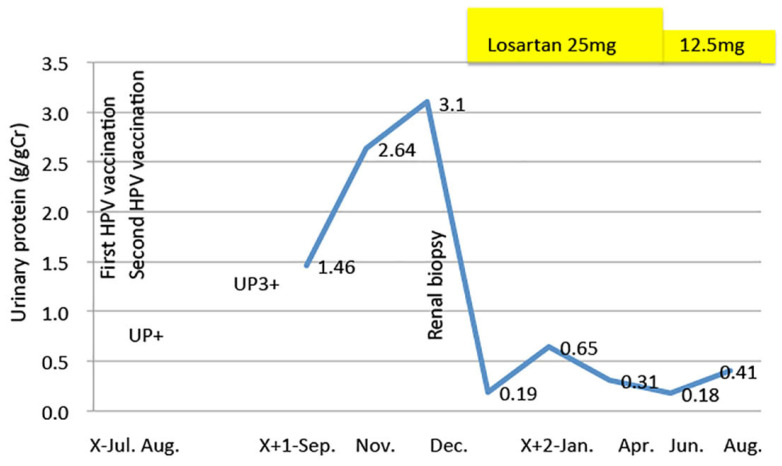
Time course of urinary protein after HPV vaccination and treatment with an angiotensin receptor blocker, losartan, as indicated by the yellow bar.

**Table 1 vaccines-10-01442-t001:** Laser microdissection mass spectrometry analysis of glomeruli of this patient and glomeruli from the 1 h renal biopsy after renal transplantation as control.

MS/MS View: 899 Proteins in 665 Clusters	Alternate ID	Control	Pt. Glm1	Pt. Glm2	Pt. Glm3	Pt. Glm4	Pt. Mean	Fold (Pt./Control)
Increased proteins								
Desmoplakin	SDP	1	198	224	201	331	239	238.5
Keratin, type I cytoskeletal 24	KRT24	26	173	169	182	176	175	6.7
Keratin, type II cytoskeletal 78	KRT78	26	154	146	143	109	138	5.3
Junction plakoglobin	JUP	nd	93	95	97	121	102	∞
luster of Keratin, type II cytoskeletal 73	KRT73	8	89	80	90	62	80	10.0
Hornerin	HRNR	nd	43	53	55	101	63	∞
Keratin, type I cytoskeletal 23	KRT23	nd	57	59	50	50	54	∞
Desmoglein-1	DSG1	0	48	49	43	70	53	∞
Calmodulin-like protein 5	CALML5	nd	42	43	39	45	42	∞
Fatty acid-binding protein 5	FABP5	nd	42	43	39	45	42	∞
Galectin-7	LGALS7	nd	20	18	22	34	24	∞
Cystatin-A	CSTA	nd	17	22	18	25	21	∞
Plakophilin-1	PKP1	1	15	18	21	36	23	22.5
Serpin B12	SERPIN	nd	15	19	12	27	18	∞
Protein-glutamine gamma-glutamyltransferase E	TGM3	nd	12	14	9	24	15	∞
Filaggrin-2	FLG2	1	10	12	6	18	12	11.5
Arginase-1	ARG1	nd	7	9	9	15	10	∞
Complement C3	C3	3	48	55	64	108	69	22.9
Cluster of Keratin, type II cytoskeletal 6A	KRT6A	236	2584	2466	2588	3111	2687	11.4
Cluster of Keratin, type I cytoskeletal 16	KRT16	158	2548	2420	2509	2754	2558	16.2
Keratin, type II cytoskeletal 1	KRT1	403	2078	1946	2049	2205	2070	5.1
Keratin, type I cytoskeletal 9	KRT9	311	1395	1269	1324	1416	1351	4.3
Deceased proteins								
Cluster of Vimentin	VIM	622	233	239	220	257	237	0.4
Cluster of Actin, cytoplasmic 2	ACTG1	500	249	230	234	227	235	0.5
Cluster of Alpha-actinin-4	ACTN4	264	84	92	80	153	102	0.4
Myosin-9	MYH9	213	78	85	93	185	110	0.5
Cluster of Tubulin beta chain	TUBB	103	36	36	34	59	41	0.4
Laminin subunit alpha-5	LAMA5	77	25	30	31	54	35	0.5
Cluster of Histone H2B type 1-M	H2BC14	101	47	40	46	61	49	0.5
basement membrane-specific heparan sulfate proteoglycan core pr.	HSPG2	26	13	14	11	46	21	0.8
Vinculin	VCL	51	13	13	15	40	20	0.4
Podocin	NPHS2	14	7	2	3	7	5	0.3
Podocalyxin	PODXL	9	5	4	4	8	5	0.6
Membranous nephropathy antigens								
Secretory phospholipase A2 receptor	PLA2R1	nd	nd	0	nd	5	1	
Thrombospondin-type -1 domain-containing 7A	THSD7A	nd	nd	nd	nd	nd	nd	
Exostosin 1 and exostosin 2	EXT1/2	nd	nd	nd	nd	nd	nd	
Protein kinase C-binding protein NELL1	NELL1	nd	nd	nd	nd	1	0.25	
Semaphorin 3b	Sema 3B	nd	nd	nd	nd	nd	nd	
Protocadherin 7	PCDH7	nd	nd	nd	nd	nd	nd	
Human high-temperature requirement A1	HTRA1	nd	nd	nd	nd	nd	nd	
Contactin 1		nd	nd	nd	nd	nd	nd	

**Table 2 vaccines-10-01442-t002:** Nephrotic syndrome and nephritis associated with vaccination.

Reference	Age Sex	Vaccine	Onset after Injection	Proteinuria	Renal Function	Renal Biopsy	Treatment	Prognosis
Patel [27]	60 F	Influenza	2 weeks	20.5 g/day	AKI	MN stage 1, AIN	HD, PSL	CR with relapse
Kutlucan [31]	56 M	Influenza	20 days	7.3 g/day	Cr 1.2 mg/dL	MN IgG, C3	PSL1 mg/kg	CR
Kao [32]	72 M	Influenza	<2 weeks	5.7 g/day	ND	ND	mPSL pulse PEX	GBS UP decreased after 10 M
Kielstein [33]	65 F	Influenza	4 days	10.8 g/day	Ccr 65 mL/min	MCNS	Conservative	CR
Gutiérrez [34]	44 M	Influenza	18 days	4 g/day	Cr 4.4 mg/dL	MCNS	PSL60 mg	CR
Mader [35]	86 F	Influenza				HSP		CR
Patel [36]	77 M	Influenza	10 days	ND	Cr2.31 mg/dL	Mesangial proliferative GN, HSP	PSL60 mg	CR
Yanai-Berar [37]	63 M	Influenza	11 days	1.5 g/day	Cr 1.8 mg/dL	Pauci-immune crescentic GN	PSL60 mg	CR
Islek [28]	4 M	HBV	8 days	2 g/m^2^/day	ND	ND	PSL	CR
Kikuchi [29]	67 F	Polyvalent pneumococcal polysaccharide	1 week	10.4 g/day	Cr 1.33 mg/dL	MCNS with TIN	mPSL pulse	CR
Claujus [38]	82 F	Tetanus–diphtheria–poliomyelitis	6 weeks	12 g/day	Cr 0.84 mg/dL	MCNS	PSL75 mg	CR
Anupama [30]	19 F	hAdOx1 nCoV-19	8 days	3.18 g/gCr	Cr1.09 mg/dL	MCNS	PSL1 mg/kg	CR
Lebedev [1]	50 M	BNT162b2 COVID-19	10 days	6.9 g/day	Cr 2.31 mg/dL	MCNS AIN	PSL	CR
Maas [39]	80 M	BNT162b2 COVID-19	7 days	15.3 g/gCr	Cr 1.43 mg/dL	MCNS	PSL80 mg	PR1
Present case	16 F	HPV	5 days	1.46 g/gCr	Cr 0.47 mg/dL	MN	ARB	PR1

## Data Availability

A table showing all of the data from the LMD-MS analysis is available upon Appendix A.

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
