# Peer review of "Juvenile Membranous Nephropathy Developed after Human Papillomavirus (HPV) Vaccination"

_vaccines, 2022, doi:10.3390/vaccines10091442_

Round 1

Reviewer 1 Report

  This paper presents some clinical results concerning possible side effects of the mRNA vaccines for SARS-CoV-2 related to HPV Infection.

 The paper is not properly addressed to vaccines, while it should go to a journal of clinical medicine. The amount of data produced by the authors appears sufficient to pose the problem, but not  to deduce some convincing conclusions. Indeed, the last section of the paper is too vague to be scientifically convincing.

Author Response

Thank you very much for evaluating our manuscript and giving us valuable comments to improve the manuscript. 

I agree with the reviewer’s comments, and shorten the discussion by omitting speculation.

Reviewer 2 Report

Review of Juvenile Membranous Nephropathy developed after Human 2 Papillomavirus (HPV) Vaccination by Haruna Arakawa, Shohei Yokoyama, Takehiro Ohira, Dedong Kang, Kazuho Honda, Yoshihiko Ueda, and Akihiro Tojo for Vaccines.
Topic: This is a single case study of a 16-year-old girl with no history of renal disease had a fever of 38 °C after her second 10 HPV vaccination and was identified as positive for proteinuria. The value of single case studies crowding an ever growing literature is questionable, but as this involves a possible vaccine response it could serve as a sentinel case for drug companies and regulators to watch for further case reports. Nevertheless, this a well planned, analysed and reported single case study.

Methods

The authors performed a renal biopsy: “a renal biopsy was performed and small spikes on PAM staining with the granular deposition of IgG1 ++ and IgG3 + on the glomerular capillary 13 wall were discovered by immunofluorescence, although PLA2R immunostaining was negative. Analysis by electron microscope showed electron density deposition in the form of fine particles under the epithelium. The diagnosis was secondary membranous nephropathy stage II. Immunostaining with the anti-p16 INK4a antibody was positive for glomerular cells, and Western blot analysis of urinary protein showed a positive band for p16 INK4a.” However, the authors failed to find further confirmation: “laser-microdissection 18 mass spectrometry analysis of a paraffin section of glomeruli failed to detect HPV proteins.”

Conclusions

The authors speculated: “It is 19 possible that the patient had already been infected with HPV and administration of the HPV vac- 20 cine, in which the transgenic virus envelope protein made into particles formed an immune complex 21 with an increase in antibody production, resulted in secondary membranous nephropathy” The authors again speculated:” After humoral immunity was induced by vaccination, an abundant antibody for HPV attacked the HPV antigen that had infected the glomerular epithelial cells and formed an in situ immune complex, resulting in membranous nephropathy. We used p16-INK4a as a surrogate marker of HPV infection [12, 13], but p16-INK4a is also detected in the glomerulus and tubules of the aging kidney and in kidneys with chronic allograft rejection [24, 45 25], even though this was not the case with this young patient.” The authors again speculated: Another possibility is that HPV envelope proteins such as E1 accumulated after vaccination in the podocytes and transformed podocytes formed membranous nephropathy. E1 proteins from HPV16 and 18 induce an overexpression of a different set of genes associated with proliferation and differentiation processes, and down regulation of immune response genes [26]. LMD-MS, in this case, showed an increase in cytoskeletal proteins and epithelial junctional proteins as well, as down regulation of nephrin, podocin, and podocalyxin may reflect HPV infection in the podocytes. Unfortunately, we could not detect HPV envelope proteins such as 53 E1, E5, E7, and E5 proteins by LMD-MS analysis

Advice to authors:

This single case study could serve as a sentinel signal to vaccine manufacturers and regulators. My initial reaction is that it is a well researched and presented case study. My subsequent reaction is that the speculations should be removed, the case thereby considerably shortened and the case presented as a POSSIBLE UNPROVEN sentinel signal.

Author Response

Answer to Reviewer 2
Topic: This is a single case study of a 16-year-old girl with no history of renal disease had a fever of 38 °C after her second 10 HPV vaccination and was identified as positive for proteinuria. The value of single case studies crowding an ever growing literature is questionable, but as this involves a possible vaccine response it could serve as a sentinel case for drug companies and regulators to watch for further case reports. Nevertheless, this a well planned, analysed and reported single case study.

Answer: Thank you very much for evaluating our manuscript carefully and giving valuable comments. We revised the manuscript according to the reviewer’s comments.

Methods

The authors performed a renal biopsy: “a renal biopsy was performed and small spikes on PAM staining with the granular deposition of IgG1 ++ and IgG3 + on the glomerular capillary wall were discovered by immunofluorescence, although PLA2R immunostaining was negative. Analysis by electron microscope showed electron density deposition in the form of fine particles under the epithelium. The diagnosis was secondary membranous nephropathy stage II. Immunostaining with the anti-p16 INK4a antibody was positive for glomerular cells, and Western blot analysis of urinary protein showed a positive band for p16 INK4a.” However, the authors failed to find further confirmation: “laser-microdissection mass spectrometry analysis of a paraffin section of glomeruli failed to detect HPV proteins.”

Answer: The primary membranous nephropathy usually showed IgG4 predominant and PLA2R is positive about 70%. In the present case did not fit with feature of primary membranous nephropathy and p16-INK4a was positive in some glomeruli, thus, we diagnosed secondary membranous nephropathy associated with HPV infection. As the laser-microdissection mass spectrometry analysis failed to detect HPV protein in the glomeruli, subepithelial immune deposits may not be directly composed with viral proteins, but immune reaction with HPV vaccine and its adjuvant. We revised discussion.

Conclusions

The authors speculated: “It is possible that the patient had already been infected with HPV and administration of the HPV vaccine, in which the transgenic virus envelope protein made into particles formed an immune complex with an increase in antibody production, resulted in secondary membranous nephropathy” The authors again speculated:” After humoral immunity was induced by vaccination, an abundant antibody for HPV attacked the HPV antigen that had infected the glomerular epithelial cells and formed an in situ immune complex, resulting in membranous nephropathy. We used p16-INK4a as a surrogate marker of HPV infection [12, 13], but p16-INK4a is also detected in the glomerulus and tubules of the aging kidney and in kidneys with chronic allograft rejection [24, 45 25], even though this was not the case with this young patient.” The authors again speculated: Another possibility is that HPV envelope proteins such as E1 accumulated after vaccination in the podocytes and transformed podocytes formed membranous nephropathy. E1 proteins from HPV16 and 18 induce an overexpression of a different set of genes associated with proliferation and differentiation processes, and down regulation of immune response genes [26]. LMD-MS, in this case, showed an increase in cytoskeletal proteins and epithelial junctional proteins as well, as down regulation of nephrin, podocin, and podocalyxin may reflect HPV infection in the podocytes. Unfortunately, we could not detect HPV envelope proteins such as 53 E1, E5, E7, and E5 proteins by LMD-MS analysis.

Answer: The reviewer is right. We have removed speculation and simply concluded that “It is possible that the patient was already infected with HPV and administration of the HPV vaccine may have caused secondary membranous nephropathy.”

Advice to authors:

This single case study could serve as a sentinel signal to vaccine manufacturers and regulators. My initial reaction is that it is a well researched and presented case study. My subsequent reaction is that the speculations should be removed, the case thereby considerably shortened and the case presented as a POSSIBLE UNPROVEN sentinel signal.

Answer: Thank you very much for the valuable comments. I agree that the speculation of immune complex formation should be delete or shorten. We have shorted the discussion and removed speculation as possible.

Round 2

Reviewer 1 Report

The authors have made some effort to avoid cinclusions out of one only case study. Now the paper cannot be improved any more. The remaining problem is if the paper can be published to a journal of vaccines or a journal of clinical medicine.  This matter needs an editorial decision.

Reviewer 2 Report

Thanks to the authors for making the requested changes